# Towards Learning to Remember in Meta Learning of Sequential Domains

## Abstract

Meta-learning has made rapid progress in past years, with recent extensions made to avoid catastrophic forgetting in the learning process, namely continual meta learning. It is desirable to generalize the meta learner's ability to continuously learn in *sequential domains*, which is largely unexplored to-date. We found through extensive empirical verification that significant improvement is needed for current continual learning techniques to be applied in the *sequential domain meta learning* setting. To tackle the problem, we adapt existing dynamic learning rate adaptation techniques to meta learn both model parameters and learning rates. Adaptation on parameters ensures good generalization performance, while adaptation on learning rates is made to avoid catastrophic forgetting of past domains. Extensive experiments on a sequence of commonly used real-domain data demonstrate the effectiveness of our proposed method, outperforming current strong baselines in continual learning. Our code is made publicly available online (anonymous) `https://github.com/ICLR20210927/Sequential-domain-meta-learning.git`.

## 1 Introduction

Humans have the ability to quickly learn new skills from a few examples, without erasing old skills. It is desirable for machine-learning models to adopt this capability when learning under changing contexts/domains, which are common scenarios for real-world problems. These tasks are easy for humans, yet pose challenges for current deep-learning models mainly due to the following two reasons: 1) Catastrophic forgetting is a well-known problem for neural networks, which are prone to drastically losing knowledge on old tasks when a domain is shifted (McCloskey & Cohen, 1989); 2) It has been a long-standing challenge to make neural networks generalize quickly from a limited amount of training data (Wang et al., 2020a). For example, the dialogue system can be trained on a sequence of domains, (hotel booking, insurance, restaurant, car services, etc) due to the sequential availability of dataset (Mi et al., 2020). For each domain, each task is defined as learning one customer-specific model (Lin et al., 2019). After finishing meta training, the model could be deployed to the previously trained domains, as the new (unseen) customers from previous domains may arrive later, they have their own (small) training data (support set) used for adapting the sequentially meta-learned models. After adaptation, the newly adapted model for the new customers can be deployed to make responses to the customers.

We formulate the above problem as sequential domain few-shot learning, where a model is required to make proper decisions based on only a few training examples while undergoing constantly changing contexts/domains. It is expected that adjustments to a new context/domain should not erase knowledge already learned from old ones. The problem consists of two key components that have been considered separately in previous research: the ability to learn from a limited amount of data, referred to as *few-shot learning*; and the ability to learn new tasks without forgetting old knowledge, known as *continual learning*. The two aspects have been proved to be particularly challenging for deep learning models, explored independently by extensive previous work (Finn et al., 2017; Snell et al., 2017; Kirkpatrick et al., 2017; Lopez-Paz & Ranzato, 2017). However, a more challenging yet useful perspective to jointly integrate the two aspects remains less explored.

Generally speaking, meta-learning targets learning from a large number of similar tasks with a limited number of training examples per class. Most existing works focus on developing the general-

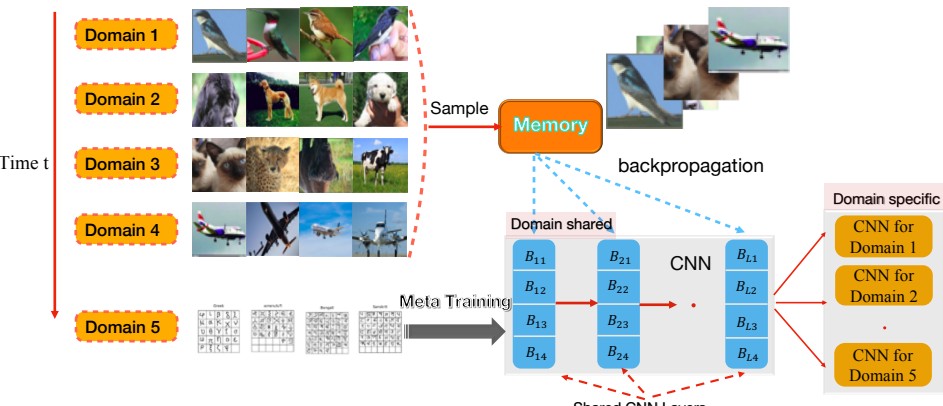

Figure 1: Meta-learning over sequential domains. Data in each domain arrive sequentially. Our model consists of a domain-shared part and a domain-specific part, all consist of a few convolutional layers (and possibly fully connected layers). The domain-shared part is shared by all domains, and each domain only owns one sub-network in the domain-specific part. Parameters (*e.g.*, convolutional filters) in each domain-shared convolutional layer $i$ (blue) are divided into $n$ blocks, denoted as $B_{i0}, B_{i1}, \cdots, B_{in}$. Each block is associated with one learnable learning rate when meta training each domain on the network. The learning rates are updated by a loss defined on the memory tasks to enforce the memorization of previous domains.

ization ability under a single context/domain (Santoro et al., 2016; Finn et al., 2017; 2018; Snell et al., 2017; Ravi & Beatson, 2019). Recently, it has been shown that catastrophic forgetting often occurs when transferring a meta-learning model to a new context (Ren et al., 2019; Yoon et al., 2020). Continual learning aims to mitigate negative backward transfer effects on learned tasks when input distribution shift occurs during sequential context changes. Related techniques of which are currently applied mostly on standard classification problems (Serrà et al., 2018; Ebrahimi et al., 2020b). In this paper, we generalize it to the sequential domain meta-learning setting, which seeks good generalization on unseen tasks from all domains with only limited training resources from previous domains. We term the problem *sequential domain meta learning*. Note this setting is different from *continual few-shot learning* that focuses on remembering previously learned low-resource tasks in a single domain. Our setting does not aim to remember on a specific task, but rather to maintain good generalization to a large amount of unseen few-shot tasks from *previous domains* without catastrophic forgetting. This setting is common and fits well in dynamic real-world scenarios such as recommendation system and dialogue training system.

The domain shift arised from this setting during meta learning poses new challenges to existing continual-learning techniques. This is mainly due to the high variability underlying a large number of dynamically formed few-shot tasks, making it infeasible for a model to explicitly remember each task. In our setting, a model is expected to remember patterns generic to a *domain*, while neglecting noise and variance of a specific few-shot *task*. This ability, termed as *remember to generalize*, allows a model to capture general patterns of a domain that repeatedly occur in batches of tasks while avoid being too sensitive to a specific few-shot task.

In this paper, we propose to address the aforementioned challenges by designing a dynamic learning-rate adaptation scheme for learning to remember previous domains. These techniques could jointly consider gradients from multiple few-shot tasks to filter out task variance and only remember patterns that are generic in each domain. Our main idea is to meta learn both the model parameters and learning rates by backpropagating both a domain loss and a memory loss to adaptively update model parameters and the learning rates, respectively. Specifically, our mechanism keeps a small memory of tasks from previous domains, which are then used to guide the dynamic and adaptive learning behaviors on different portions of the network parameters. The proposed mechanism is versatile and applicable to both the metric-based prototypical network (Snell et al., 2017) and the gradient-based ANIL (Raghu et al., 2020) meta-learning model.

Our contributions are summarized as follows:

- We propose a challenging benchmark that requires a meta learning model to sequentially learn on a sequence of domains enduring domain shift without much forgetting on previous domains.

- We extend meta learning models with existing dynamic learning rate modeling techniques. This can mitigate catastrophic forgetting through meta learning both model parameters and learning rates to dynamically control the network update process. This can be seamlessly integrated into both metric-based and gradient-based meta learning approaches.

- We conduct extensive experiments on multiple public datasets under different sequential domain few-shot learning scenarios. We further test functionality of the dynamic learning-rate update mechanism for both metric-based and gradient-based meta-learning approaches. Comparisons are made towards a wide range of representative continual-learning techniques and models. Results demonstrate that our method outperforms strong baselines by a large margin.

## 2 RELATED WORKS

### 2.1 META LEARNING

Meta learning (Schmidhuber, 1993), aka, learning to learn, aims to rapidly adapt to a new task by reusing previous experience through training on a large number of tasks. Meta learning can be roughly classified into the following categories: 1) Metric/Embedding-based approaches such as (Vinyals et al., 2016; Snell et al., 2017; Edwards & Storkey, 2017), which map input data into embedding (feature) spaces with decisions made based on some distance metric in the feature space; 2) Black-box learning methods such as (Andrychowicz et al., 2016; Graves et al., 2014; Mishra et al., 2018); 3) Optimization-based methods such as (Finn et al., 2017; Ravi & Larochelle, 2017; Li et al., 2017; Antoniou & Storkey, 2019), which improve gradient-based optimization algorithms or learn to initialize network parameters; and 4) Bayesian meta-learning methods such as (Ravi & Beatson, 2019; Finn et al., 2018; Yoon et al., 2018b; Grant et al., 2018; Wang et al., 2020b). These methods are used to either interprete and understand MAML (Grant et al., 2018), or to model uncertainty of meta learning models (Yoon et al., 2018b; Finn et al., 2018; Wang et al., 2020b). 5) Memory-based meta learning (Santoro et al., 2016; Munkhdalai & Yu, 2017; Mikulik et al., 2020), which apply additional memory component for meta learning. Online meta learning (Finn et al., 2019) is also related to us. They focus on forward transfer, i.e., achieving better performance on future task and use all the data from previous tasks to do meta learning, while our setting is significantly different from theirs as we focus on mitigating catastrophic forgetting with only very limited access to previous domains.

Dynamically updating the learning rates for networks is not new and has been explored in several contexts. Meta-SGD (Li et al., 2017) learns the per parameter learning rates for meta learning to improve flexibility and performance. (Gupta et al., 2020) use dynamic learning rates to mitigate forgetting in online continual learning. T-net (Lee & Choi, 2018) learns a metric in activation space, which informs the update direction and step size for task-specific learning. Flennerhag et al. (2020) proposes e warped gradient descent to meta-learns an efficiently parameterised preconditioning matrix to dynamically update the network. Our work extends dynamic learning rate techniques to sequential domain meta learning setting to mitigate catastrophic forgetting.

### 2.2 CONTINUAL LEARNING

Continual learning tackles the problem of maintaining knowledge when input distribution shift happens in sequentially arriving tasks. There are different methods to address this problem, including 1) retaining memory for future replay (Lopez-Paz & Ranzato, 2017; Chaudhry et al., 2019a; Riemer et al., 2019; Chaudhry et al., 2019b); 2) designing tailored network architectures (Rusu et al., 2016; Fernando et al., 2017; Yoon et al., 2018a); 3) performing proper regularization during parameter updates (Kirkpatrick et al., 2017; Zenke et al., 2017; von Oswald et al., 2019); and 4) introducing Bayesian methods for model parameter inference (Nguyen et al., 2018; Ebrahimi et al., 2020a). Specifically, methods based on memory replay store representative samples from old tasks and rehearsal is performed during training (Lopez-Paz & Ranzato, 2017; Chaudhry et al., 2019a; Riemer et al., 2019). Recent research also utilizes generative models to memorize previously seen data (Lesort et al., 2019). Representatives of architecture-based methods include Progressive Neural Networks (Rusu et al., 2016), PathNet (Fernando et al., 2017), Dynamically Expandable Networks (Yoon et al., 2018a), Hard Attention Mask (HAT) (Serrà et al., 2018) and PackNet (Mallya & Lazeb-

nik, 2017), *etc*. These models explicitly modify network topology to preserve previous knowledge. The classic architecture-based approaches proposed in (Serrà et al., 2018) and (Mallya & Lazebnik, 2017) do not fit into this setting, as they attempt to fully remember each historic task. Progressive Neural Networks (Rusu et al., 2016) guarantee zero forgetting but at the cost of growing network architectures and increasing parameters rapidly, which is unaffordable in memory-constraint cases. Regularization-based methods constrain the updated parameters to avoid drastic changes to previously learned tasks (Kirkpatrick et al., 2017; Zenke et al., 2017; von Oswald et al., 2019; Ebrahimi et al., 2020c). They can restrict the capacity to meta-learning of new domains, thus it could hurt the performance on a new domain. Bayesian-based methods model parameters in a probabilistic way, and then parameters are updated either based on their posterior distributions (Nguyen et al., 2018) or on their uncertainty (Ebrahimi et al., 2020a). However, in the context of meta-learning, the uncertainty or posterior estimation could be highly inaccurate due to the *small-data* setting in each task, thus hindering the performance. Recently, there are works using meta learning to improve continual learning. For example, (Javed & White, 2019) proposes to learn versatile representations by explicit training towards minimizing forgetting.

### 2.3 MULTI-DOMAIN META LEARNING

As a new research direction, multi-domain meta learning aims to achieve good generalization across multiple domains. (Triantafillou et al., 2020) releases a dataset containing few-shot tasks from multiple domains. (Tseng et al., 2020) proposes to use transformation layers to associate feature distributions across different domains. (Vuorio et al., 2019) proposes a tailored initialization process of model parameters based on task embedding to enable proper functionality in multiple domains. It is worth noting that the aforementioned multi-domain few-shot learning methods assume data from all domains are jointly available during training. We consider a more challenging setting where domain shift comes sequentially, thus a model needs to remember knowledge from previous domains.

### 2.4 INCREMENTAL FEW-SHOT LEARNING

Incremental few-shot learning (Gidaris & Komodakis, 2018; Ren et al., 2019; Yoon et al., 2020) aims to handle new categories with limited resources while preserving knowledge on old categories. The task requires building a generalized model while preventing catastrophic forgetting, with an implicit assumption of unlimited access to the base categories. This paper, by contrast, focuses on the case where only very limited access to previous domain is available, and generalization to *unseen* categories in previous domains is required. (Gidaris & Komodakis, 2018) introduces a novel cosine similarity function to relate incremental classes with base classes. (Ren et al., 2019) proposes attention attractor network to regularize the learning of new categories. (Yoon et al., 2020) proposes to integrate two sets of features with one from a pretrained base module and the other from a meta-trained module for a smooth adaptation on novel categories. For incremental few-shot learning, the increasing new classes are assumed to be in the same domain as the base classes, which is different from the changing domains in our setting.

### 2.5 CONTINUAL FEW-SHOT LEARNING

Continual few-shot learning is a relatively new research topic. (Antoniou et al., 2020) proposes a general framework for various continual few-shot learning settings, and proposes benchmarks for proper performance evaluation. (Caccia et al., 2020) introduces *Continual-MAML* for online fast adaptation to new tasks while accumulating knowledge on old tasks and conducted experiments on three domains, our setting is different from theirs because they assume previous tasks can be revisited, we assume only very limited access to previous domains. (Jerfel et al., 2019) considers task distribution shift within a single domain and proposes Dirichlet process mixture of hierarchical Bayesian models (DPM) to tackle the problem. Their ideas are interesting, but they assume different set of parameters for each cluster, which is not efficient in memory-limited setting.

## 3 THE PROPOSED METHOD

**Problem setup** Our goal is to perform meta learning on a sequence of $N$ domains, denoted as $\mathcal{D}_1, \mathcal{D}_2, \ldots, \mathcal{D}_N$. Each domain $\mathcal{D}_i$ consists of data divided into meta-training, meta-validation and meta-testing sets, denoted as $\{\mathcal{D}_i^{tr}, \mathcal{D}_i^{val}, \mathcal{D}_i^{test}\}$. Only $\mathcal{D}_i^{tr}$ is used for training a meta-learning model; $\mathcal{D}_i^{val}$ is used for validation; and $\mathcal{D}_i^{test}$ is used for meta testing. Each task $\mathcal{T} \in \mathcal{D}_i$ consists of $K$ data examples, $\{(\boldsymbol{x}^k, \boldsymbol{y}^k)\}_{k=1}^K$. A task is divided into $\mathcal{T}^{tr}$ and $\mathcal{T}^{test}$. Our goal is to sequentially meta-learn a model $f_{\boldsymbol{\theta}}$ that maps inputs $\boldsymbol{x}$ to $\boldsymbol{y}$ for each arriving domain while not forgetting too much of all previous domains. To this end, our framework consists of a very limited memory of

$\mathcal{M}_d$, which is used to store a small number of training tasks from each previous domain $\mathcal{D}_d$. In domain $\mathcal{D}_q$, the parameters are adopted from that of the last domain, $\boldsymbol{\theta}_{q-1}$. Data available for training are thus $\mathcal{D}_q \bigcup (\bigcup_{d=1}^{q-1} \mathcal{M}_d)$. Our goal is to update the parameter to $\boldsymbol{\theta}_q$ such that it achieves good performance on a domain $\mathcal{D}_q$ by transferring useful knowledge from previous domains without degrading performance on these domains. The final performance is evaluated on the meta-test sets of current and all previous domains.

**Basic idea and model architecture**  For effective learning without catastrophic forgetting of previous domains, our basic ideas include 1) minimizing a loss on the current domain to achieve fast adaptation and good generalization; 2) adjusting the adaptation via optimizing the learning rates to force the model to remember previous domains. In our implementation, we consider a convolutional neural network (CNN) based classifier, which is supposed to evolve over sequences of domains via meta-learning adaptation. We assume all domains share the same CNN-based structure for feature extraction, while the model should also have the flexibility to define domain-specific components. As a result, we divide the CNN classifier into two parts, corresponding to bottom convolutional layers for feature extraction and top convolutional layers for classification. The bottom layers of the CNN are defined as domain-shared layers to serve as a common feature extractor for all domains. In addition, the top layers of the CNN are defined as domain-specific layers, which are defined as $N$ parallel sub-CNNs with each accessible by only one domain. The network structure is illustrated in Figure 1. For more flexible adaptation, we group the parameters (the convolutional filters) in each domain-shared layer into several blocks and associate each block with one learning rate. In other words, different filters will be updated with different learning rates, which are also adaptively optimized to enforce the memorization of previous domains (specified in the following sections). The CNN structure is also reflected in Figure 1. Note the learning rate in each of the domain-specific layers is not optimized as the layers in different domains are independent with each other, making the adaptive learning-rate scheme used in the domain-shared part inapplicable.

**Problem formulation**  Similar to (Javed & White, 2019), we treat the domain-shared and domain-specific parameters separately. The meta parameters $\boldsymbol{\theta}^S$ (parameters of bottom convolutional layers) and its corresponding network block learning rates $\boldsymbol{\lambda}^S$ are shared across all the domains. We use $\boldsymbol{\theta}_{1:T}^{\mathcal{D}}$ to denote the domain-specific parameters and $\boldsymbol{\lambda}_{1:T}^{\mathcal{D}}$ denote the domain-specific learning rates from domain $1 \cdots T$, where each element $\boldsymbol{\theta}_q^{\mathcal{D}}$ and $\boldsymbol{\lambda}_q^{\mathcal{D}}$ represent domain $\mathcal{D}_q$ specific parameters and its corresponding learning rates. All the learnable parameters for domain $\mathcal{D}_q$ are defined as $\boldsymbol{\theta}_q = \{\boldsymbol{\theta}^S, \boldsymbol{\theta}_q^{\mathcal{D}}\}$ and learning rates as $\boldsymbol{\lambda}_q = \{\boldsymbol{\lambda}^S, \boldsymbol{\lambda}_q^{\mathcal{D}}\}$. $\boldsymbol{\lambda}^S$ are dynamically updated and $\boldsymbol{\lambda}_{1:T}^{\mathcal{D}}$ are fixed during the meta-training process. During meta testing, $\boldsymbol{\lambda}^S$ and $\boldsymbol{\lambda}_{1:T}^{\mathcal{D}}$ are not used and updated. We use $\boldsymbol{\theta}_{q,j}$ to denote all the learnable parameters at $j^{th}$ iteration and $\boldsymbol{\lambda}_{q,j}^S$ to denote all learnable learning rates at the $j^{th}$ iteration when meta training in domain $\mathcal{D}_q$. Each element of $\boldsymbol{\lambda}_{q,j}^S$ is the learning rate for parameters inside one block, each of which consists of several filters. Note that $\boldsymbol{\lambda}^S$ and $\boldsymbol{\lambda}_{q,j}^S$ are the same set of learning rates, the difference is that the latter one denotes the specific learning rates at one iteration. $\mathcal{M} = \bigcup_{d=1}^{q-1} \mathcal{M}_d$ denotes all the memory tasks in all the domains before domain $\mathcal{D}_q$.

$\mathcal{L}(\mathcal{T}, \boldsymbol{\theta}(\boldsymbol{\lambda}))$ is the task-specific loss function with model parameters $\boldsymbol{\theta}$ that depends on learning rates $\boldsymbol{\lambda}$ and data from task $\mathcal{T}$. $\mathcal{J}_{\mathcal{D}_q^{tr}}(\boldsymbol{\theta}_{q,j+k}(\boldsymbol{\lambda}_{q,j}^S))$ (Equation 5) represents the expected loss function with respect to training task distributions in domain $\mathcal{D}_q$ and is used for updating model parameters. $\mathcal{J}_{\mathcal{M}}(\boldsymbol{\theta}_{q,j+K}(\boldsymbol{\lambda}_{q,j}^S))$ (Equation 2) is the expected loss function with respect to memory task distributions and is used for adjusting learning rates. Detailed definitions are as following.

$$\min_{\boldsymbol{\lambda}_{q,j}^S} \mathcal{J}_{\mathcal{M}}(\boldsymbol{\theta}_{q,j+K}(\boldsymbol{\lambda}_{q,j}^S)) \tag{1}$$

$$\mathcal{J}_{\mathcal{M}}(\boldsymbol{\theta}_{q,j+K}(\boldsymbol{\lambda}_{q,j}^S)) = \underset{\mathcal{T}_i \in \mathcal{M}}{\mathbb{E}} \mathcal{L}(\mathcal{T}_i, \boldsymbol{\theta}_{q,j+K}(\boldsymbol{\lambda}_{q,j}^S)) \tag{2}$$

$$\boldsymbol{\theta}_{q,j+k+1}(\boldsymbol{\lambda}_{q,j}^S) = \boldsymbol{\phi}(\boldsymbol{\theta}_{q,j+k}, \boldsymbol{\lambda}_{q,j}; \mathcal{D}_q^{tr}), \text{ for } k = 0, 1, 2, \ldots, K-1, \tag{3}$$

$$\boldsymbol{\phi}(\boldsymbol{\theta}_{q,j+k}, \boldsymbol{\lambda}_{q,j}; \mathcal{D}_q^{tr}) = \boldsymbol{\theta}_{q,j+k} - \boldsymbol{\lambda}_{q,j} \nabla_{\boldsymbol{\theta}_q} \mathcal{J}_{\mathcal{D}_q^{tr}}(\boldsymbol{\theta}_{q,j+k}(\boldsymbol{\lambda}_{q,j})) \tag{4}$$

$$\mathcal{J}_{\mathcal{D}_q^{tr}}(\boldsymbol{\theta}_{q,j+k}(\boldsymbol{\lambda}_{q,j}^S)) = \underset{\mathcal{T}_i \in \mathcal{D}_q^{tr}}{\mathbb{E}} \mathcal{L}(\mathcal{T}_i, \boldsymbol{\theta}_{q,j+k}(\boldsymbol{\lambda}_{q,j}^S)) \tag{5}$$

$\boldsymbol{\theta}_{q,j+k+1}(\boldsymbol{\lambda}_{q,j}^S)$ is the function of $\boldsymbol{\lambda}_{q,j}^S$ instead of $\boldsymbol{\lambda}_{q,j}$ since $\boldsymbol{\lambda}_q^{\mathcal{D}}$ is fixed during training $\mathcal{D}_q$ for the reasons stated above. Where $K$ is the number of iterations in domain $\mathcal{D}_q$ after current iteration, we use $K = 1$ for simplicity. $\boldsymbol{\phi}(\cdot, \cdot)$ denotes a gradient adaptation function. Similarly, for the learning rate adaptation at domain $\mathcal{D}_q$, based on the objective equation 2, an adaptation process is defined as:

$$\boldsymbol{\lambda}_{q,j+1}^S = \boldsymbol{\lambda}_{q,j}^S - \eta \nabla_{\boldsymbol{\lambda}^S} \mathcal{J}_{\mathcal{M}}(\boldsymbol{\theta}_{q,j+K}(\boldsymbol{\lambda}_{q,j}^S)) , \tag{6}$$

where $\eta$ is the hyper learning rate of the learning-rate update rule.

To sum up, our method is a bi-level adaptation process. The lower level optimizes on the current domain, while the upper level optimizes on previous domains for updating the block-wise learning rates. Detailed algorithms describing the meta-training and testing phases are given in Algorithm 1 and 2 (Appendix A), respectively.

**Domain-level meta loss computation** Our model contains a component to train on previous tasks stored in the memory to optimize the learning rates. In practice, only a minibatch of data is used at each iteration. Note each task is in the format of training-validation pairs from one domain. If one simply uses one task from one domain at each iteration, the model could overfit to this domain and may not generalize well to other domains. To alleviate this issue, we propose to compute the meta loss at two different levels, called intra-domain and inter-domain meta losses. The intra-domain meta loss is defined with tasks in the same domain to ensure good generalization in one domain. The inter-domain meta loss, by contrast, is defined with tasks sampled across all available domains in the memory to encourage model adaptation across different domains. When calculating the inter-domain loss with one task from the memory, at each iteration, we augment the data with those from tasks of other domains. In this way, the model can be trained to better generalize to other domains. Figure 2 illustrates our proposed method. More details can be found in Appendix A.

---

**Algorithm 1** Meta training.

**Require:** A sequence of training domain data $\mathcal{D}_1, \mathcal{D}_2, \ldots, \mathcal{D}_N$; Initial learning rates $\boldsymbol{\lambda}^S$ and $\boldsymbol{\lambda}_{1:T}^{\mathcal{D}}$ to $\boldsymbol{\lambda}_0$ and model parameters $\boldsymbol{\theta}^S = \boldsymbol{\theta}_0$; $\boldsymbol{\lambda}_{1:T}^{\mathcal{D}}$ is fixed during the training process.

**Require:** Maximum number of iterations $M$ for each domain.

1: $\mathcal{M} = \{\}$
2: **for** $q = 1$ to $N$ **do**
3:     Randomly initialize $\boldsymbol{\theta}_q^{\mathcal{D}}$
4:     **for** $j = 1$ to $M$ **do**
5:         Sample batch of tasks $\mathcal{T}$ from $P(\mathcal{D}_q)$, distribution over tasks in domain $\mathcal{D}_q$
6:         Evaluate $\mathcal{J}_{\mathcal{D}_q^{tr}}(\boldsymbol{\theta}_{q,j+1}(\boldsymbol{\lambda}_{q,j}^S)) = \mathbb{E}_{\mathcal{T} \in \mathcal{D}_q^{tr}} \mathcal{L}(\mathcal{T}, \boldsymbol{\theta}_{q,j+1}(\boldsymbol{\lambda}_{q,j}^S))$
7:         Perform adaptation: $\boldsymbol{\theta}_{q,j+1} = \boldsymbol{\theta}_{q,j} - \boldsymbol{\lambda}_{q,j} \nabla_{\boldsymbol{\theta}_q} \mathcal{J}_{\mathcal{D}_q^{tr}}(\boldsymbol{\theta}_{q,j}, \boldsymbol{\lambda}_{q,j})$
8:         **if** $q = 1$ **then**
9:             $\boldsymbol{\lambda}_{q,j}^S = \boldsymbol{\lambda}_0$
10:        **else**
11:          $\boldsymbol{\lambda}_{q,j+1}^S = \boldsymbol{\lambda}_{q,j}^S - \eta \nabla_{\boldsymbol{\lambda}^S} \mathcal{J}_{\mathcal{M}}(\boldsymbol{\theta}_{q,j}, \boldsymbol{\lambda}_{q,j}^S)$
12:        **end if**
13:     **end for**
14:     $\mathcal{M}_q = \{$a small batch of sampled tasks from$\mathcal{D}_q\}$

15:     $\mathcal{M} = \mathcal{M} \cup \mathcal{M}_q$
16: **end for**

---

## 4 EXPERIMENT

We compare our method against several related strong baselines on a sequence of five domain datasets, which exhibit large domain shift, thus posing new challenges for existing methods. We evaluate the effectivenss of our method on both gradient-based and metric-based meta-learning frameworks. We also conduct ablation studies to verify the effectiveness of each component of our model. Our implementation is based on Torchmeta (Deleu et al., 2019). Results are reported in terms of mean and standard deviation over three independent runs.

**Implementation details** For prototypical-network-based (Protonet) (Snell et al., 2017) baselines, we use a four-layer CNN with 64 filters of kernel size being 3 as shared domain feature extractor for all the domains. The last layer is defined as domain specific with 64 filters. No fully connected layers are used following existing works. For gradient-based meta-learning algorithms (Raghu et al., 2020), following (Antoniou et al., 2019), we use a three-layer CNN with 48 filters of kernel size being 3 as shared domain feature extractor, and one convolutional layer with 48 filters and one fully-connected layer for domain-specific learning. Such architectures for Prototypical network and

ANIL are widely adopted in existing works on continual learning to mitigate catastrophic forgetting (Serrà et al., 2018; Ebrahimi et al., 2020b). 750 evaluation tasks from each domain are used for meta testing. The hyper learning rate $\eta$ is set to 1e-4 for Protonet and 5e-5 for ANIL. The gradients with respect to all the learnable learning rates are clipped to [-10, 10] for both Protonet and ANIL. Unless otherwise specified, the number of learning blocks for each CNN layer is 4, the number of memory tasks for each previous domain is 6 and the arriving order of the domain sequence is MiniImagenet, CIFARFS, Omniglot, Aircraft and CUB. More implementation details are given in Appendix A.

**Datasets** The five datasets include Miniimagenet (Vinyals et al., 2016), CIFARFS (Bertinetto et al., 2019), Omniglot (Lake et al., 2011), CUB (Welinder et al., 2010), AIRCRAFT (Maji et al., 2013). All images are resized into $84 \times 84$. We follow the splits in torchmeta* for the Miniimagenet, CIFARFS, Omniglot and CUB datasets. For AIRCRAFT, we follow the split in (Vuorio et al., 2019). We compare all the baselines on 5-way-1-shot and 5-way-5-shot learning settings.

**Evaluation metrics** We adapt existing work on continual learning to use the evaluation metrics of ACC and BWT (Ebrahimi et al., 2020b). ACC is defined as the average meta testing classification accuracy across all domains (the higher the better), and BWT measures the average forgetting on all the previous domains evaluated at the end of the sequential meta learning task (the lower the better). Formaly, the ACC and BWT are defined as: $\text{ACC} = \frac{1}{N} \sum_{j=1}^{N} a_{N,j}$ and $\text{BWT} = \frac{1}{N-1} \sum_{i=1}^{N-1} a_{N,i} - a_{i,i}$, where $a_{n,i}$ is the meta testing accuracy on domain $i$ after meta training on domain $n$, $a_{N,i} - a_{i,i}$ measures the forgetting score on domain $i$ after meta training on domain $N$.

**Baselines** For meta-learning methods, we compare our method with ANIL (Raghu et al., 2020) and Prototypical Network (Snell et al., 2017). The former is a simplified version of MAML (Finn et al., 2017) with inner loop only conducted on the final layer. For continual learning methods, we compare our method with related strong baselines, which are regularization based method such as the Elastic Weight Consolidation (Kirkpatrick et al., 2017), architecture-based methods (Hard Attention Mask (HAT) (Serrà et al., 2018) with publicly available code from the authors), Bayesian methods (Ebrahimi et al., 2020b) (with the authors' implementation), Memory-based (including A-GEM (Chaudhry et al., 2019a), ER-Ringbuffer (Chaudhry et al., 2019b) and Meta Experience Replay (MER) (Riemer et al., 2019)). We also learn all the domains jointly in a multi-domain meta-learning setting, whose performance can be considered as the upper bound for our sequential learning setting. We also provide a simple baseline, which trains sequentially without using any external memory. We denote this method as "Sequential", whose performance can be considered as reference for the catastrophic forgetting of our setting. It is worth noting that the above baselines have only been used for traditional continual learning. We adapt these methods to our setting to demonstrate the ineffectiveness of these existing techniques when generalizing them to a meta-learning setting.

**Comparisons to baselines** Table 1 show the comparisons to the baselines we constructed for this problem setting in terms of 5-way 5-shot accuracy. Results of 5-way 1-shot classification is given in Appendix A. In the table, 'N/A' means the BWT is not available since the method does not learn sequentially. Our method significantly outperforms baselines. Especially, in the Protonet-based model, the performance of our model almost matches that of the joint-training model, indicating excellent memorization of past domains. From the results of "Sequential" baseline, we see that there is a significant performance drop if no external mechanism is introduced to prevent forgetting. Among the baselines, *ER-Ringbuffer* seems to perform worse overall. We believe this is because of the repeatedly-trained memory, which leads to overfitting, making recovery of previous knowledge difficult. Memory data in our method are not directly fit into network but are only used for guiding the training of network on current domain data, thus avoiding overfitting. The inferior performance of *UCB* might be because the uncertainty estimation with limited data is inaccurate in the meta-learning setting. *HAT* is also worse because it simply remembers history tasks but not domain general information. Furthermore, *A-GEM* restricts gradient updates on new examples in the same direction as the gradient direction in memory tasks, potentially leading to wrong update directions. In *MER*, the fast weight for each task in memory across different domains could vary significantly, making updates by Reptile oscillate and unstable, and thus hindering its performance.

**Sensitivity to domain ordering** We study the sensitivity of all methods on the order of domain arrivals. We use a different domain-sequence order as: CIFARFS, MiniImagenet, Aircraft, CUB and Omniglot. Results of 5-way-5-shot learning are summarized in Table 2. Appendix A details the

---

*https://github.com/tristandeleu/pytorch-meta

Table 1: Compare to Baselines

| Algorithm | 5 Way 5 Shots | | Algorithm | 5 Way 5 Shots | |
|---|---|---|---|---|---|
| | ACC | BWT | | ACC | BWT |
| Protonet-Sequential | $54.74 \pm 0.11$ | $-24.68 \pm 0.15$ | ANIL-Sequential | $50.68 \pm 0.25$ | $-27.54 \pm 0.34$ |
| Protonet-EWC | $59.38 \pm 0.42$ | $-11.15 \pm 0.51$ | ANIL-EWC | $48.18 \pm 0.24$ | $-30.24 \pm 0.26$ |
| Protonet-HAT | $62.58 \pm 0.24$ | $-8.96 \pm 0.30$ | ANIL-HAT | $47.02 \pm 0.26$ | $-30.73 \pm 0.32$ |
| Protonet-UCB | $57.82 \pm 0.05$ | $-13.81 \pm 0.09$ | ANIL-UCB | $51.58 \pm 0.17$ | $-25.96 \pm 0.23$ |
| Protonet-A-GEM | $57.38 \pm 0.32$ | $-22.09 \pm 0.26$ | ANIL-A-GEM | $51.56 \pm 0.38$ | $-26.45 \pm 0.40$ |
| Protonet-ER-Ringbuffer | $58.25 \pm 0.31$ | $-19.06 \pm 0.36$ | ANIL-ER-Ringbuffer | $45.14 \pm 0.25$ | $-35.29 \pm 0.28$ |
| Protonet-MER | $60.79 \pm 0.16$ | $-11.96 \pm 0.15$ | ANIL-MER | $51.50 \pm 0.23$ | $-25.67 \pm 0.26$ |
| Protonet-Ours | $\mathbf{68.72 \pm 0.22}$ | $\mathbf{-3.22 \pm 0.17}$ | ANIL-Ours | $\mathbf{56.62 \pm 0.32}$ | $\mathbf{-15.28 \pm 0.40}$ |
| Joint-training | $71.81 \pm 0.29$ | N/A | Joint-training | $73.52 \pm 0.20$ | N/A |

results of 5-way-1-shot setting. It can be seen that although there are some performance differences compared to those of the previous order, our method still outperforms the baselines.

Table 2: Compare to Baselines with different domain ordering

| Algorithm | 5 Way 5 Shots | | Algorithm | 5 Way 5 Shots | |
|---|---|---|---|---|---|
| | ACC | BWT | | ACC | BWT |
| Protonet-Sequential | $51.28 \pm 0.31$ | $-28.79 \pm 0.39$ | ANIL-Sequential | $44.18 \pm 0.39$ | $-33.79 \pm 0.35$ |
| Protonet-EWC | $59.12 \pm 0.35$ | $-14.87 \pm 0.27$ | ANIL-EWC | $46.27 \pm 0.37$ | $-32.24 \pm 0.40$ |
| Protonet-HAT | $60.61 \pm 0.28$ | $-10.89 \pm 0.15$ | ANIL-HAT | $42.52 \pm 0.15$ | $-34.79 \pm 0.31$ |
| Protonet-UCB | $50.36 \pm 0.18$ | $-20.85 \pm 0.23$ | ANIL-UCB | $45.32 \pm 0.25$ | $-31.28 \pm 0.36$ |
| Protonet-A-GEM | $59.75 \pm 0.28$ | $-17.85 \pm 0.35$ | ANIL-A-GEM | $45.87 \pm 0.32$ | $-32.01 \pm 0.41$ |
| Protonet-ER-Ringbuffer | $61.52 \pm 0.30$ | $-14.21 \pm 0.40$ | ANIL-ER-Ringbuffer | $41.35 \pm 0.30$ | $-37.82 \pm 0.37$ |
| Protonet-MER | $62.71 \pm 0.18$ | $-11.23 \pm 0.10$ | ANIL-MER | $47.19 \pm 0.23$ | $-28.45 \pm 0.29$ |
| Protonet-Ours | $\mathbf{67.29 \pm 0.32}$ | $\mathbf{-4.12 \pm 0.30}$ | ANIL-Our | $\mathbf{55.89 \pm 0.22}$ | $\mathbf{-12.36 \pm 0.31}$ |
| Joint-training | $71.81 \pm 0.29$ | N/A | Joint-training | $73.52 \pm 0.20$ | N/A |

We use another different domain-sequence order as: Omniglot, Aircraft, CUB, CIFARFS and Mini-Imagenet. Since Omniglot is used as first domain, the model cannot learn good representations for the following domains, thus poses more challenges than the other two domain sequences. Table 3 shows the results, where the baseline *Protonet-fixfirst* is the method that freeze the model parameters after finishing training on the first domain. For this baseline, there is small BWT, this is because of random testing task variance at different time. In this case, all the baselines and our methods performance drop significantly compared to the sequence with CIFARFS or Miniimagenet as first domain. It indicates that this domain sequence is significantly more challenging than the other two domain sequences.

**Effect of domain-level meta loss** This sets of experiments verify the effectiveness of our proposed domain-level meta loss in Section 3. We compare the domain-level meta loss with the standard intra-domain meta loss, which only considers task data in one domain. The results are shown in Table 4. It is clear that our proposed domain-level meta loss consistently outperforms the intra-domain meta loss by obtaining better accuracy.

Table 3: Compare to Baselines with Omniglot as first training domain

| Algorithm | 5 Way 1 Shot | | 5 Way 5 Shots | |
|---|---|---|---|---|
| | ACC | BWT | ACC | BWT |
| Protonet-Sequential | $42.69 \pm 0.26$ | $-22.18 \pm 0.32$ | $57.92 \pm 0.21$ | $-18.32 \pm 0.19$ |
| Protonet-EWC | $37.98 \pm 0.19$ | $-19.43 \pm 0.22$ | $54.02 \pm 0.18$ | $-13.24 \pm 0.23$ |
| Protonet-HAT | $41.02 \pm 0.12$ | $-18.55 \pm 0.17$ | $58.35 \pm 0.21$ | $-16.14 \pm 0.15$ |
| Protonet-UCB | $41.32 \pm 0.20$ | $-18.01 \pm 0.32$ | $58.73 \pm 0.28$ | $-17.28 \pm 0.21$ |
| Protonet-A-GEM | $43.18 \pm 0.27$ | $-21.10 \pm 0.21$ | $59.96 \pm 0.16$ | $-14.86 \pm 0.18$ |
| Protonet-ER-Ringbuffer | $41.92 \pm 0.28$ | $-22.54 \pm 0.29$ | $59.35 \pm 0.23$ | $-16.96 \pm 0.20$ |
| Protonet-MER | $39.60 \pm 0.28$ | $-24.21 \pm 0.31$ | $58.10 \pm 0.22$ | $-16.26 \pm 0.17$ |
| Protonet-fixfirst | $44.29 \pm 0.19$ | $-0.16 \pm 0.14$ | $56.29 \pm 0.29$ | $-0.22 \pm 0.12$ |
| Protonet-Ours | $\mathbf{46.56 \pm 0.18}$ | $\mathbf{-14.34 \pm 0.25}$ | $\mathbf{62.68 \pm 0.15}$ | $\mathbf{-11.12 \pm 0.25}$ |
| Joint-training | $53.92 \pm 0.36$ | N/A | $71.81 \pm 0.29$ | N/A |

Table 4: Effect of domain-level meta loss

| Algorithm | 5 Way 1 Shot | | 5 Way 5 Shots | |
|---|---|---|---|---|
| | ACC | BWT | ACC | BWT |
| Protonet-intra and inter meta loss | $53.36 \pm 0.27$ | $-3.39 \pm 0.15$ | $68.72 \pm 0.22$ | $-3.22 \pm 0.17$ |
| Protonet-intra loss | $52.29 \pm 0.31$ | $-4.02 \pm 0.18$ | $67.69 \pm 0.21$ | $-3.97 \pm 0.23$ |
| ANIL-intra and inter meta loss | $45.85 \pm 0.22$ | $-10.19 \pm 0.27$ | $56.62 \pm 0.32$ | $-15.28 \pm 0.40$ |
| ANIL-intra loss | $44.16 \pm 0.30$ | $-11.38 \pm 0.20$ | $55.10 \pm 0.19$ | $-16.97 \pm 0.33$ |

Table 5: Compare to our method with different number of learning blocks

| Algorithm | 5 Way 1 Shot | | 5 Way 5 Shots | |
|---|---|---|---|---|
| | ACC | BWT | ACC | BWT |
| Protonet-Ours-2block | $50.12 \pm 0.19$ | $-7.15 \pm 0.12$ | $65.25 \pm 0.23$ | $-6.64 \pm 0.29$ |
| Protonet-Ours-4block | $53.36 \pm 0.27$ | $-3.39 \pm 0.15$ | $68.72 \pm 0.22$ | $-3.22 \pm 0.17$ |
| Protonet-Ours-8block | $53.51 \pm 0.33$ | $-3.21 \pm 0.55$ | $68.82 \pm 0.33$ | $-3.01 \pm 0.21$ |

Table 6: Effect of number of memory tasks. Top, memory with one task for each domain; Bottom, memory with six tasks for each domain

| Algorithm | 5 Way 1 Shot | | 5 Way 5 Shots | |
|---|---|---|---|---|
| | ACC | BWT | ACC | BWT |
| Protonet-A-GEM | $38.50 \pm 0.09$ | $-29.37 \pm 0.17$ | $55.22 \pm 0.11$ | $-24.01 \pm 0.12$ |
| Protonet-ER-Ringbuffer | $39.41 \pm 0.16$ | $-27.41 \pm 0.26$ | $56.17 \pm 0.28$ | $-23.46 \pm 0.25$ |
| Protonet-MER | $39.04 \pm 0.19$ | $-25.73 \pm 0.25$ | $58.37 \pm 0.24$ | $-17.52 \pm 0.31$ |
| Protonet-Ours | $50.73 \pm 0.17$ | $-5.56 \pm 0.32$ | $66.92 \pm 0.29$ | $-4.96 \pm 0.30$ |
| Protonet-A-GEM | $40.07 \pm 0.40$ | $-26.47 \pm 0.46$ | $57.38 \pm 0.32$ | $-22.09 \pm 0.26$ |
| Protonet-ER-Ringbuffer | $44.12 \pm 0.19$ | $-21.83 \pm 0.17$ | $58.25 \pm 0.31$ | $-19.06 \pm 0.36$ |
| Protonet-MER | $46.72 \pm 0.25$ | $-14.82 \pm 0.11$ | $60.79 \pm 0.16$ | $-11.96 \pm 0.15$ |
| Protonet-Ours | $\mathbf{53.36 \pm 0.27}$ | $\mathbf{-3.39 \pm 0.15}$ | $\mathbf{68.72 \pm 0.22}$ | $\mathbf{-3.22 \pm 0.17}$ |

**Effect of number of learning block**   Our method assigns a different learning rate for each block of the network parameters. A finer grained division of learning blocks allows more flexible control, but leads to higher computation cost. In this experiment, we investigate the impact of blocks numbers to model performance. Specifically, we implement each layer with 2, 4 and 8 blocks. Results are shown in Table 5. It can be seen that when the block number is relatively large (4 in our case), the accuracies are relatively stable. Thus we can simply set the block number to 4 in practice.

**Effect of a small memory size**   We compare performance differences when the number of tasks stored in the memory for each domain is small. Table 6 shows the results of 1 and 6 tasks from each domain. It is interesting to see that with a small memory to store only one task from each domain, performances of all the memory-based baselines drop significantly, while our method maintains relatively stable performance. This might be due to the fact that with a smaller memory size, all memory-based baselines either overfit to the small memory or are impacted by unstable gradient direction updates, which would not happen in our method.

## 5   CONCLUSION

In this paper, we propose a challenging Benchmark that requires a meta learning model to sequentially meta learn on a sequence of domains with domain shift but without much forgetting on previous domains. Then, we extend existing dynamic learning rate techniques to existing meta learning model to meta learn the learning rates on meta parameters blocks, which can be seamlessly integrated into both metric-based and gradient-based meta learning approaches to mitigate catastrophic forgetting. The adaptation on parameters maintains generalization performance on current domain, while adaptation on learning rates is made remember knowledge of past domains. Our proposed method significantly outperforms existing continual-learning techniques adapted to our problem, achieving significantly better performance than strong baselines. There are several possible future directions to be pursued for future work, such as the cases of imbalanced classes in each task, scaling to longer domain sequences, and fewer training tasks in each arrival domain.

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

## A APPENDIX

**Joint-training details**    We describe details of ANIL and Prototypical Network, the two base models tested in our framework.

*For ANIL*: The meta batch size for each domain is set to 3. For 1-shot and 5-shot learning, the number of training data for each task are set to 1 and 5, and the number of query data is 15 for both cases. The number of inner loop update steps is 5 with a learning rate of 1e-2. The outer-loop learning rate is 1e-3, and the number of meta training iterations is 20000 with the Adam optimizer (Kingma & Ba, 2014).

*For Prototypical Network*: The meta batch size for each domain is 3, and the number of meta training iterations is 40000. For 1 shot learning, the number of support data for each task is 1, the number of query data for each task is 10. For 5 shot learning, the number of support data for each task is 5, the number of query data for each task is 10. The meta training loss is optimized by Adam (Kingma & Ba, 2014), and the learning rate is 1e-3.

**Sequential domain training details**

For ANIL and Prototypical Network, we adopt the following hyperparameter.

- **ANIL** (Raghu et al., 2020)

  The meta batch size for each domain is 3. The total number of meta training iterations is 40000. The number of meta training iterations for each domain is 8000. For 1-shot and 5-shot learning, the number of training data for each task are set to 1 and 5, and the number of query data is 15 for both cases. The number of inner loop update steps is 5, the inner loop learning rate is 1e-2, the outer loop learning rate is 1e-3, the outer loop meta training loss is optimized by Adam (Kingma & Ba, 2014).

- **Prototypical Network** (Snell et al., 2017)

  the meta batch size for each domain is 3, and the number of meta training iterations is 40000. The number of meta training iterations for each domain is 8000. For 1-shot and

5-shot learning, the number of training data for each task are set to 1 and 5, and the number of query data is 10 for both cases. The meta training loss is optimized by Adam (Kingma & Ba, 2014), and the learning rate is 1e-3.

For continual learning method, we adopt the following hyperparameter

- **HAT details** (Serrà et al., 2018)

  This baseline is adapted from authors implementation [†]. The output $h_l$ of the units in layer $l$ is element-wise multiplied by the following: $h'_l = a^t_l \odot h_l$, where $a^t_l$ is the annealed version of the single layer gated domain embedding $e^t_l$, defined as

  $$a^t_l = \sigma(s e^t_l) \tag{7}$$

  Other hyperparameter are following (Serrà et al., 2018), the stability parameter or scaling parameter $s$ is annealed according to

  $$s = \frac{1}{s_{max}} + (s_{max} - \frac{1}{s_{max}}) \frac{b-1}{B-1} \, , \tag{8}$$

  where $s_{max} = 400$, $b$ is $1, \cdots, B$ is batch index and $B$ is the total number of batches.

- **EWC details** (Kirkpatrick et al., 2017)

  We vary the weight penalty $\lambda$ for avoiding drastic change in parameters with $\lambda = 5, 1 \times 10^1, 5 \times 10^1, 1 \times 10^2, 5 \times 10^2, 1 \times 10^3, 5 \times 10^3$ and select the best as our baseline.

- **A-GEM details** (Chaudhry et al., 2019a)

  The implementation is based on [‡].

- **MER details** (Riemer et al., 2019)

  The implementation is based on [§]. The within batch meta-learning rate $\beta$ is set to 0.03, across batch meta-learning rate $\gamma$ to 1.0, and batches per example to 5.

- **UCB details** (Ebrahimi et al., 2020a)

  Following (Ebrahimi et al., 2020a), we scale the learning rate of $\mu$ and $\rho$ for each parameter proportional to its importance $\Omega$, which is measured by its variance, to reduce changes in important parameters and adapt authors implementation [¶]. We use 10 Monte Carlo samples at each iteration as in (Ebrahimi et al., 2020a).

- **ER-Ringbuffer details** (Chaudhry et al., 2019b)

  We use a fixed amount of memory of each previous domain to jointly train with current domain.

Other meta learning related settings are the same as those in sequential domain meta training as described above.

**Domain-level meta loss** In this part, we will elaborate on the computation of proposed domain-level meta loss. To alleviate the overfitting issue of a specific domain during the learning process, we propose to compute the meta loss at two different separated levels, intra-domain and inter-domain meta losses. The intra-domain meta loss is defined with tasks in the same domain to ensure good generalization in one domain. The inter-domain meta loss, by contrast, is defined with tasks sampled across all available domains in the memory to encourage model adaptation across different domains. When calculating the inter-domain loss with one task from the memory, at each iteration, we augment the data with those from tasks of other domains. In this way, the model can be trained to better generalize to other domains.

---

[†] https://github.com/joansj/hat
[‡] https://github.com/GMvandeVen/continual-learning
[§] https://github.com/mattriemer/MER
[¶] https://github.com/SaynaEbrahimi/UCB

We use $S_p^u$ to denote the examples labeled with $u$ for domain $p$, and $S_q^v$ to denote the examples labeled with class $v$ for domain $q$. Suppose $P$-way $K$-shot learning is performed in both domain $p$ and domain $q$. At each iteration, the two tasks from both domains are combined together, and the problem becomes $2P$-way $K$-shot learning. That is, the network is used for $2P$ class classification problem for images in both domain $p$ and domain $q$

Define the prototype for class $u$ in a domain as

$$c^u = \frac{1}{|S^u|} \sum_{(x_p, y_p) \in S^u} f_{\theta}(x_p), u \in \{0, 1, \cdots, 2P - 1\} . \quad (9)$$

This will encourage the network to differentiate the image classes across two different domains and help the network generalize across different domains.

For a given query data $x$, the goal is to model

$$p_{\theta}(y = u | x) = \frac{exp(-d(f_{\theta}(x), c^u))}{\sum\limits_{u'=0}^{2P-1} exp(-d(f_{\theta}(x), c^{u'}))} \quad (10)$$

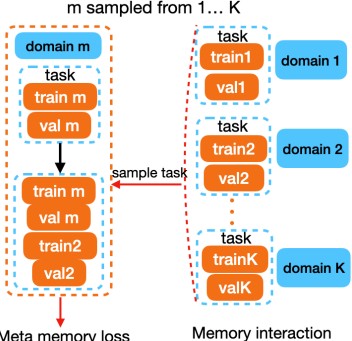

Figure 2: The meta memory loss mechanism. For one task in domain $q$ from the memory pool , we randomly sample the data from some tasks of other domains and combine them with their own data to calculate the task-specific loss.

The memory loss is defined as minimizing the log probability $\mathcal{J}(\theta) = -\log p_{\theta}(y = u | x)$, where $d$ is distance function.

For MAML (ANIL), let $\mathcal{D}_q^{val} \bigcup \mathcal{D}_p^{val}$ denote the validation data from both domain $q$ and $p$. The loss is defined as

$$\underset{\theta}{\arg\min} \, \mathcal{L}^{meta}(\theta, \phi^*(\theta), \mathcal{D}_q^{val} \bigcup \mathcal{D}_p^{val}) \quad (11)$$

$$\phi^*(\theta) = \underset{\phi}{\arg\min} \, \mathcal{L}^{task}(\theta, \phi, \mathcal{D}_q^{tr}) \quad (12)$$

Both mechanisms for the prototypical network and MAML (ANIL) can avoid overfitting and generalize across different domains.

**On learning-rate optimization** Learning rate is one of the most important and effective hyperparameter in standard neural-network optimization (Baydin et al., 2018; Franceschi et al., 2017; Donini et al., 2020) and continual learning (Ebrahimi et al., 2020b). Different from standard supervised learning whose goal is to minimize the generalization error, continual learning aims to avoid catastrophic forgetting while maintaining good performance on current domain. Inspired by standard gradient-based hyperparameter optimization algorithms such as (Baydin et al., 2018; Franceschi et al., 2017; Donini et al., 2020), we develop a novel automatic and adaptive learning-rate update scheme for newly arrival domains. Our goal is achieved by updating the learning rate for each parameter block, which is designed to adaptively update parts of the network to fit future domains while remembering previous domains.

**More Results**

**Compare to Baselines**

**5-way 1-shot learning** Table 7 show the comparisons to the baselines we constructed for this problem setting in terms of 5-way 1-shot accuracy. In the table, 'N/A' means the BWT is not available since the method does not learn sequentially. Our method significantly outperform baselines. Especially, similar to the performance in 5-way 5-shot learning in Table 1, in the Protonet-based model, the performance of our model almost matches that of the joint-training model, indicating excellent memorization of past domains.

**Compare to Baselines with different domain ordering**

Table 7: Compare to Baselines (5-way 1-shot)

| Algorithm | 5 Way 1 Shot | | Algorithm | 5 Way 1 Shot | |
| | ACC | BWT | | ACC | BWT |
| --- | --- | --- | --- | --- | --- |
| Protonet-Sequential | $36.31 \pm 0.18$ | $-30.97 \pm 0.12$ | ANIL-Sequential | $37.55 \pm 0.21$ | $-25.12 \pm 0.26$ |
| Protonet-EWC | $42.21 \pm 0.21$ | $-19.06 \pm 0.36$ | ANIL-EWC | $38.86 \pm 0.22$ | $-24.15 \pm 0.31$ |
| Protonet-HAT | $49.98 \pm 0.19$ | $-6.67 \pm 0.24$ | ANIL-HAT | $39.90 \pm 0.11$ | $-20.02 \pm 0.15$ |
| Protonet-UCB | $41.26 \pm 0.17$ | $-9.46 \pm 0.07$ | ANIL-UCB | $38.86 \pm 0.23$ | $-21.35 \pm 0.29$ |
| Protonet-A-GEM | $40.07 \pm 0.40$ | $-26.47 \pm 0.46$ | ANIL-A-GEM | $34.62 \pm 0.36$ | $-29.04 \pm 0.42$ |
| Protonet-ER-Ringbuffer | $44.12 \pm 0.19$ | $-21.83 \pm 0.17$ | ANIL-ER-Ringbuffer | $31.49 \pm 0.47$ | $-32.45 \pm 0.56$ |
| Protonet-MER | $46.72 \pm 0.25$ | $-14.82 \pm 0.11$ | ANIL-MER | $41.40 \pm 0.19$ | $-19.24 \pm 0.21$ |
| Protonet-Ours | $\mathbf{53.36 \pm 0.27}$ | $\mathbf{-3.39 \pm 0.15}$ | ANIL-Ours | $\mathbf{45.85 \pm 0.22}$ | $\mathbf{-10.19 \pm 0.27}$ |
| Joint-training | $53.92 \pm 0.36$ | N/A | Joint-training | $57.18 \pm 0.38$ | N/A |

**5-way 1-shot learning** We study the sensitivity of all methods on the order of domain arrivals in the setting of 5-way 1-shot learning. We use a different domain-sequence order as: CIFARFS, MiniImagenet, Aircraft, CUB and Omniglot. Results are summarized in Table 8. Similar to the results in Table 2, it can be seen that although there are some performance differences compared to those of the previous order, our method consistently outperforms the baselines.

Table 8: Compare to Baselines with different domain ordering (5-way 1-shot)

| Algorithm | 5 Way 1 Shot | | Algorithm | 5 Way 1 Shot | |
| | ACC | BWT | | ACC | BWT |
| --- | --- | --- | --- | --- | --- |
| Protonet-Sequential | $42.66 \pm 0.14$ | $-23.04 \pm 0.12$ | ANIL-Sequential | $39.61 \pm 0.20$ | $-20.95 \pm 0.22$ |
| Protonet-EWC | $43.97 \pm 0.20$ | $-17.36 \pm 0.33$ | ANIL-EWC | $39.18 \pm 0.29$ | $-22.12 \pm 0.35$ |
| Protonet-HAT | $48.92 \pm 0.41$ | $-9.18 \pm 0.26$ | ANIL-HAT | $39.02 \pm 0.12$ | $-22.62 \pm 0.10$ |
| Protonet-UCB | $39.64 \pm 0.14$ | $-10.83 \pm 0.09$ | ANIL-UCB | $39.97 \pm 0.37$ | $-19.28 \pm 0.41$ |
| Protonet-A-GEM | $46.81 \pm 0.23$ | $-18.78 \pm 0.35$ | ANIL-A-GEM | $37.76 \pm 0.35$ | $-24.91 \pm 0.27$ |
| Protonet-ER-Ringbuffer | $48.60 \pm 0.35$ | $-14.85 \pm 0.18$ | ANIL-ER-Ringbuffer | $35.42 \pm 0.31$ | $-26.03 \pm 0.37$ |
| Protonet-MER | $49.92 \pm 0.33$ | $-9.06 \pm 0.32$ | ANIL-MER | $40.86 \pm 0.37$ | $-18.34 \pm 0.31$ |
| Protonet-Ours | $\mathbf{52.17 \pm 0.12}$ | $\mathbf{-0.74 \pm 0.19}$ | ANIL-Ours | $\mathbf{45.25 \pm 0.20}$ | $\mathbf{-6.02 \pm 0.28}$ |
| Joint-training | $53.92 \pm 0.36$ | N/A | Joint-training | $57.18 \pm 0.38$ | N/A |

**Meta testing of current and all previous domains** Algorithm 2 shows the algorithm of meta testing the model on current and all the previous domains.

---

**Algorithm 2** Meta testing.

---

**Require:** A sequence of training domain data $\mathcal{D}_1, \mathcal{D}_2, \ldots, \mathcal{D}_N$;
**Require:** Learned meta parameter $\boldsymbol{\theta}$ of the models after meta training on domain $N$.
1: **for** $q = 1$ to $N$ **do**
2:     Sample $T$ meta-testing tasks $\mathcal{T}_q^{test}$ from $P(\mathcal{D}_q)$, the distribution over tasks in domain $\mathcal{D}_q$
3:     Evaluate the meta testing accuracy and measurements of forgetting for domain $\mathcal{D}_q$ using the meta learned model $f(\mathcal{T}, \boldsymbol{\theta}_N^M), \mathcal{T} \in \mathcal{T}_q^{test}$
4: **end for**
5: Evaluate the average meta testing accuracy on all learned domains and measurements of forgetting over all previous domains

---

