# OpenReview forum: "Towards Learning to Remember in Meta Learning of Sequential Domains"
_ICLR.cc/2021/Conference — Reject_

### Official Review · AnonReviewer1 · 2020-10-25
**potentially interesting idea but needs more work**

**Rating:** 5
**Confidence:** 5

**Review:**

The authors propose a new instantiation of Continual Few-Shot learning (CFSL) with multiple domains coined sequential domain meta-learning.
They also propose an extension of the meta-SGD [1] algorithm to CFSL where the learning rates are learned on tasks from past domains to alleviate forgetting.
Both the setting and the idea have potential, but the paper still needs some work and considerable weaknesses.
Mainly, the setting needs further motivation and clarifications, and the method needs some justification as well as stronger performance.

On the setting:
- 1.1 The setting's motivations are self-driving car- and dialogue system-like scenarios. The authors claim that these are aligned with the newly proposed setting. In these examples, however, the hypothetical methods would need to adapt to the changing data distribution in an **online** manner and without an explicit support set. In the new setting, methods are tested at the end of training on all past tasks, i.e. long **after** the methods were in the tested domains and with a different model. When a new setting is proposed, it should be clearly motivated. I suggest the authors either improve the current motivations or adapt their settings to better align with real-life continual-learning scenarios. For the latter, the authors could, e.g., report online cumulative performance on all domains with domain revisits, similarly to [2].

- 1.2 It seems like the methods have access to the task label to properly chose their output head (or domain-dependent parameters)? If this is so, the methods are operating in the task-aware setting, where forgetting can be solved by freezing and growing the network. In that case, the authors should add this as a baseline. If they cannot beat it, they should show on what metrics they can outperform it, e.g. maybe their method is more computationally efficient. Also, task-aware doesn't seem to be aligned with the proposed motivations.

- 1.3 The meta-test protocol (ALgorithm 2) is unclear. Are the fast weights retrained? If so, how? Are the learned learning rates $\lambda_q$ ever used? If not, why are they learned?



On the method:
- 2.1 The authors rush to explain their double adaptation method but their mechanism is not justified anywhere. Specifically, why would you learn the learning rate on past tasks? and why would you backprop through $\theta_q^j$ which is adapted to domain $q$?

- 2.2 The authors should acknowledge that learning the learning rate (or double adaptation) is a well-known trick now. They should explain how they extend this method and change the tone of the text such that their proposed method seems less novel.

- Going back to 1.3, in ALgo 2, are the learned learning rates ever used? Because algorithm 2 only uses $\theta_N^M$ i.e the model adapted to domain $N$.

On the experiment:

- 3.1 why didn't the authors run a hyper-parameter search?

- 3.2 Table 3 ablates their proposed double adaptation on past domain scheme, which is the novel part of their method. The gains are however relatively small. Furthermore, with the large number of hyper-parameters and without a hyper-parameter search, the gains could be noise introduced by the authors iterating on their method by not on the baselines. I suggest the authors run a hyperparameter sensitivity analysis.

- 3.3 The authors explain a new memory selection mechanism that is unclear and doesn't work well yet. I encourage the authors to remove that section and Table 6, or significantly improve it. Also, on "We then divide the losses into Q clusters with kmeans. ":  if the losses are scalars, why are you using k-means?



Other concerns:
- figure 2 seems unconstructive. I suggest the authors remove it or move it to the Appendix.

- put lines in the algorithms.

- The introduction is hard to follow. Also, the 4th paragraph should be moved and merged into the Related Work section.

- In the related work,  the authors claim that [2] is part of a group of papers that operates in a single domain. However, this is not true for [2] which operates in multiple domains within one experiment, e.g. the Omniglot / MNIST / Fashion-MNIST experiment. Also, [3] is a Meta-Continual learning method and not an incremental few-shot learning one.

typos:
- in abstract and elsewhere: "catastrophe" forgetting--> "catastrophic"




_____

**POST REBUTTAL**

The authors have provided some clarifications. I suggest they use them to improve the paper.
I'm increasing my score to a 5, thus still not in favor of acceptance.
_______

[1] Zhenguo Li, Fengwei Zhou, F. Chen, and H. Li.  Meta-sgd:  Learning to learn quickly for few shot learning. ArXiv, abs/1707.09835, 2017.

[2] Massimo Caccia, P. Rodr ́ıguez, O. Ostapenko, Fabrice Normandin, Min Lin, L. Caccia, Issam H.Laradji, I. Rish, Alexandre Lacoste, D. V ́azquez, and Laurent Charlin.  Online fast adaptation and knowledge accumulation: a new approach to continual learning.ArXiv, abs/2003.05856, 2020.

[3] Khurram   Javed   and   Martha   White.Meta-learning   representations   for   continual   learn-ing.InAdvances   in   Neural   Information   Processing   Systems   32,    pp.   1820–1830.CurranAssociates,Inc.,2019.URLhttp://papers.nips.cc/paper/8458-meta-learning-representations-for-continual-learning.pdf.

---

> ### Author Response · Authors · 2020-11-23
> **Response to Reviewer 1 (Part 1)**
>
> Thank you for your constructive feedback and we improved our paper based on your helpful comments. We addressed your concerns and provided responses below
>
> Q: On the setting with specifIc scenario and motivation.
>
> A: We acknowledge that self driving cars and dialogue systems need to make decisions in real time, but given the complexity of these systems, extensive training on numerous domains before testing is still necessary for current machine learning models to achieve satisfactory performance during testing in these applications. For example, auto driving companies need to train their models under different scenarios on millions to billions miles of data before testing. For dialogue systems, [1] provide examples of sequentially learning a sequence of dialogue domains. After training, the system could be deployed to previously trained domains instead of only testing on online tasks. Personalized customer model for each customer in each domain can be viewed as a single task with their own training dataset (support set) [3].
>
> For example, the dialogue system can be trained on a sequence of domains, (hotel booking, insurance, restaurant, car services, etc) due to the sequential availability of dataset [1]. For each domain, each task is defined as learning one customer-specific model [3].  After finishing meta training, the model could be deployed to the previously trained domains, as the new (unseen) customers from previous domains may arrive later, they have their own (small) training data (support set) used for adapting the sequentially meta-learned models.  After adaptation, the newly adapted model for the new customers can be deployed to make responses to the customers.
>
> Above examples means the system does not need to online adapt to new data, they can also adapt to tasks from previously trained domains. For the above scenario, the domain-aware labels are important for newly arrived customer, as newly arrived customers may have fewer training data and given domain-aware label will be helpful for tuning  which domain-specific model for providing proper learned model weights to generate domain-aware responses.  We believe this example is well aligned with the proposed setting and our method.
>
> [1] Continual Learning for Natural Language Generation in Task-oriented Dialog Systems. Fei Mi, Liangwei Chen, Mengjie Zhao, Minlie Huang, Boi Faltings. EMNLP 2020
> [2] Domain Adaptive Dialog Generation via Meta Learning. Kun Qian, Zhou Yu. ACL 2019
> [3] Personalizing Dialogue Agents via Meta-Learning. Zhaojiang Lin, Andrea Madotto, Chien-Sheng Wu, Pascale Fung. ACL 2019
>
>
> Q:  forgetting can be solved by freezing and growing the network?
>
> A:  This is  true only when the model can learn good representations on the first domain, such as Miniimagenet.  We use another different domain-sequence with Omniglot as first domain, the domain order is: Omniglot, Aircraft, CUB, CIFARFS and Mini-Imagenet.  Since Omniglot is used as first domain, the model cannot learn good representations for the following domains, thus poses more challenges than the other two domain sequences.  See Table 3 for the results, where the baseline Protonet-fixfirst is the method that freeze the model parameters after finishing training on the first domain.  In this case, even though all the baselines and our methods performance drop significantly compared to the sequence with CIFARFS or Miniimagenet as first domain, our method still outperforms all baselines. Overall speaking, considering the different domain sequences, if we do not know the domain sequences beforehand, dynamic learning rates are indeed necessary.
>
> Q: the meta-test protocol (ALgorithm 2) is unclear. fast weights and the learned learning rates are used?
>
> A: The learned meta parameters are not changed after finishing meta training. Note our method is used for learning meta parameters, instead of task-specific parameter. It does not need retraining after finishing the meta training process. The learned learning rates are not used in meta-testing. They are only used to adjust the shared (across all the arriving domains) meta parameters  learning speed during meta training, i.e., it helps  to avoid the model to update too fast on meta parameters that are important for past domains. During meta testing, e.g. for prototypical network, we simply use the meta learned prototypical network parameters adapted to the last domain to do few shot testing on all the domains.
>
>
>
> Q: why didn't the authors run a hyper-parameter search? suggest the authors run a hyperparameter sensitivity analysis.
>
> A: Our method does not have much hyperparameter, only has hyper-learning rate (for changing the learning rate of parameters), number of blocks per convolutional layer, the memory size.  We already present the results of sensitivity with respect to number of blocks and memory size, see table 5 and 6. The learning rate typically chosen between 1e-4 and 1e-6 works well.

---

> > ### Author Response · Authors · 2020-11-23
> > **Response to Reviewer 1 (Part 2)**
> >
> > Q: 4th paragraph merge with related work
> >
> > We merged them with related work
> >
> > Q: related work discussion issues
> >
> > A: We fixed reference issues. We have corrected the reference paper descriptions. See the related work discussion (red part)
> >
> > Q: figure 2 seems unconstructive.
> >
> > A: We moved to appendix
> >
> > Q: memory selection mechanism
> >
> > A: We removed that part
> >
> > Q: put lines in the algorithms.
> >
> > A: We put lines in algorithms now

---

### Official Review · AnonReviewer4 · 2020-10-28
**An interesting contribution that requires some clarifications**

**Rating:** 6
**Confidence:** 5

**Review:**

Summary

At the heart of this paper are two separate contributions. The first is a new online meta-learning problem setting where the meta-learner acts on a sequence of few-shot learning *domains*, as opposed to tasks within a single domain. The second is a method for meta-learning with this form of domain shift.

As far as I know, this is the first benchmark that tackles online meta-learning over few-shot domains. This is a much needed push in the direction of obtaining harder continual learning benchmark - I'm quite excited about this line of work and would encourage the authors to continue to pursue this line of work and include longer sequences of domains to truly test the continual capabilities of our current methods.

The authors also propose a meta-learner for the newly introduced problem setup. This meta-learner makes use of two well-known concepts form the literature, experience replay and learning-rate adaptation. In particular, the model's parameters are adapted to the current task (few-shot) using meta-learned learning rates. These learning rates are meta-learned over all tasks in the memory in a multi-task loss function.

Pros:
- A new benchmark that is considerably more challenging than previous online meta-learning benchmarks.
- Extensive benchmarking of a wide range of baselines from both meta-learning and continual learning.
- Strong performance from the proposed method.

Cons:
- Unclear claims to novelty.
- Confusing writing at critical steps.

Recommendation: reject

Motivation:
I believe the proposed benchmark would be of service to the community and should eventually be published at a peer-review venue, however the current manuscript contains critical issues that prevent me from recommending acceptance. I'm open to changing my score should the authors address my concerns below.

Main concerns:

- *Contributions:* The authors list three contributions. Of these, the first relates to both adapting parameters and learning rates and the second relates to using layer-wise meta-learned learning rates. Both are well known approaches that have been explored extensively in the literature. Meta-learning learning rates was proposed in [1] and have since been explored in a variety of contexts. Adapting both layers and learning rates have also been explored in a variety of forms in episodic meta-learning (e.g. [2]). Further, recent meta-learners [e.g. 3, 4] learn model parameters for fast task adaptation and meta-learn optimiser parameters to avoid catastrophic forgetting, precisely what is claim as a contribution here. This is not to say that there are no algorithmic contributions in this paper, but rather that it is impossible to tease out what the authors claim as their own contribution to the field.

- *Technical correctness:* The writing becomes very dense on pages 4 and 5, where the method is introduced, making it hard to understand precisely what the authors propose. In particular, they describe their method as a bi-level optimisation problem, but Eqs. 1-4 do not support this description.  Both losses have the same input arguments (they only differ in the expectation over tasks) and hence represent a multi-task setup. This is not a well defined problem because the learning rates have no effect on the loss unless the gradient update is taken into account. Similarly, the authors use a single \theta to denote model parameters, but in Figure 1 and in the text mention that they treat the lower layers of the CNN differently from the top (task-specific layers), as in [5]. This is not described in the algorithm, where it appears as if all model parameters are being tuned to the current task and all layer-learning rates are being meta-learned on replays of previous tasks.

Minor concerns:

- *Clarity:* the manuscript would benefit from a simpler and clearer presentation of the problem setup. At the end of the day, the paper proposes to meta-learn over a sequence of few-shot learning domains, as opposed to a sequence of tasks from the same domain. This will require a higher degree of generalisation from meta-learners, and should stress our current methods. There is no need to make the motivation more intricate than that.  Similarly for the proposed method, a simpler and more direct presentation would greatly help the reader understand what is being proposed and what is being borrowed from previous works. I'm still unclear as to what 'double adaptation' means. Is it just that both learning rates and model parameters are being tuned?

- *Citations:* while the related work section contains a wealth of citations, the introduction makes sweeping claims (such as "it has been shown that catastrophic forgetting often occurs when transferring a meta-learning model to a new context") that should be substantiated with appropriate citations. It is also a bit unfair to invalidate all previous works as not applicable to this problem setup because of "high variability underlying a large number of dynamically formed few-shot tasks". This is an empirical matter and not a an absolute fact.

- *Experiments:* generally, the experimental section is sound and features a broad set of baselines and conducts a battery of ablations. While this is great, the one ablation I was hoping for was to see what happens if you change the order of the domains such that miniImagenet and CIFAR are at the end. In general, having one (or both, as in this case) as the first two tasks means the meta-learner can learn good representations for downstream tasks pretty much immediately. This seems to precisely counter-act the goal of introducing a harder benchmark. Finally, I'm a little unclear as to whether catastrophic forgetting is measured 0-shot without re-adapation, or few-shot by allowing adaptation to tasks from past domains given \theta^j_q?

Typos:
- catastrophe forgetting -> catastrophic forgetting
- scarifying ->degrading(?)

Post Rebuttal

The authors have improved the context of their work and clarified their proposed method. While the technical novelty is somewhat limited, the proposed method does well and the benchmark introduced herein should be of interest to the community. As such I have increased my score.

References
[1] Li et. al. 2017. Meta-SGD: Learning to Learn Quickly for Few-Shot Learning.
[2] Lee et. al. 2018. Gradient-Based Meta-Learning with Learned Layerwise Metric and Subspace.
[3] Flennerhag et. al. 2019. Meta-Learning with Warped Gradient Descent.
[4] Gupta et. al. 2020. La-MAML: Look-ahead Meta Learning for Continual Learning.
[5] Javed et. al. 2019. Meta-Learning Representations for Continual Learning.

---

> ### Author Response · Authors · 2020-11-23
> **Response to Reviewer 4**
>
> Thanks a lot for your insightful comments. Your feedback improves our paper a lot.  We summarized your concerns and addressed them below.
>
> Q: statement of contribution:
>
> A: Thank you for pointing out these related works. We read these papers and they are indeed closely related to our work. We revised our statement of contributions in the following. Also see the paper revision in the contribution statement in the introduction.
>
> [1] We propose a challenging benchmark that requires a meta learning model to sequentially meta learn on a sequence of domains with domain shift but without much forgetting on previous domains.
>
> [2] We extend meta learning models with existing dynamic learning rate modeling techniques. This can mitigate catastrophic forgetting through meta learning both model parameters and learning rates to dynamically control the network update process. This can be seamlessly integrated into both metric-based and gradient-based meta learning approaches.
>
> Q: Technical correctness about notations and objective function
>
> A: We rewrite and clarify the method part. We discuss them separately and use different notations about domain-shared and domain-specific parameters and learning rates. We reformulate the optimization objective function and adaptation process. You can see the updated version in the problem formulation part of our revision.
>
> Q: presentation clarity
>
> A: We have made revision about method statement and use direct presentation of proposed method
>
> Q: citations in the introduction to support the claims.
>
> A: We added more citations to support the claims in the introduction part.
>
> Q: statement of all previous works as not applicable to this problem setup
>
> A: We have made revision to this statement. See the introduction (red part).
>
>
> Q: Experiment on different ordering without good representation on first domains,  the domain ordering such that miniImagenet and CIFAR are at the end.
>
> A: This is a good question. We did another set of experiments using a different domain sequence: Omniglot-Aircraft-CUB-CIFARFS-MiniImagenet. Results are shown in Table 3 (see the text that is marked as red for experiment descriptions). In this experiment, our method still outperforms baselines, the performance of almost all the baselines and our method also drops significantly. This means this sequence is more challenging than the other two.
>
>
> Q: unclear as to whether catastrophic forgetting is measured 0-shot without re-adapation, or few-shot by allowing adaptation to tasks from past
>
> A: This is measured by few-shot testing for each task. For prototypical networks, there is no parameter adaptation. For MAML(ANIL), adaptation is done by gradient descent.  Since the testing tasks from all the domains are unseen and testing tasks are performed on unseen categories, class labels are randomly generated, a small number of labeled examples are necessary to help the model predict the label of testing examples.
>
> Q: Typos
>
> A: We have fixed typos.

---

### Official Review · AnonReviewer3 · 2020-10-29
**Needs some clarity**

**Rating:** 5
**Confidence:** 3

**Review:**

This work focuses on sequential adaptation of a model without forgetting. Their goal is to minimize the catastrophic interference of the model while learning a new few-shot task coming from a different domain. To that end, they introduce a problem setup where the model receives a sequence of few-shot tasks from different domains.

I am not sure I entirely understand that the motivation and the proposed algorithm in this work. Overall the algorithm resembles replay-based continual learning approaches. Also, the algorithm feels similar to Online Meta-Learning. How does adaptation of learning rate address catastrophic interference? Please take a look at my comments below. Maybe, it is helpful to provide a real task for this setup. Having that said, I appreciate the authors' effort put into evaluating multiple baselines.

Comments and questions:
-What is the value of N (i.e how many tasks were shown to the model?)? I am curious about different runs with varied N, for each model including baselines.
-Maybe discuss memory and accuracy trade-off? Some baselines don’t require a replay buffer nor iterative re-training.
-Please provide references on the following statements - Most existing works focus on developing the generalization ability under a single context/domain. Recently, it has been shown that catastrophic forgetting often occurs when transferring a meta-learning model to a new context.
-How does the proposed setup and algorithm compare to those of “Finn, C. Online Meta-Learning”?
-Some missing references: “Schmidhuber, J. A neural network that embeds its own meta-levels.”, “Santoro, A. Meta-Learning with Memory-Augmented Neural Networks”, “Munkhdalai, T. Meta Networks,” etc.
-Please note that miniImageNet subset was introduced in Matching Nets.
-The first sentence in Section 2.1 seems to describe an application of meta-learning to few-shot learning rather than meta-learning itself. There is also a line of works on memory-based meta-learning (Mikulik, V. Meta-trained agents implement Bayes-optimal agents).

---

> ### Author Response · Authors · 2020-11-23
> **Response to Reviewer 3**
>
> Thanks a lot for your helpful comments. We revised our paper and provided our responses to your question.
>
> Q: how many tasks were shown to the model? the value of N
>
> A: N is the number of domains. We set it to be 5. For example, the domains are: Omniglot-Aircraft-CUB-CIFARFS-MiniImagenet. Each dataset is simulated as one domain. For varied N, the longer the domain sequence is, the problem becomes more challenging. For shorter domain sequences, the problem becomes easier to solve. We will add more experiments on longer domain sequences in future work.
>
>
> Q: missing references
>
> A: Thank you for pointing out these related papers. Finn, C. Online meta learning focuses on forward transfer and uses all the previous task data to do meta learning, thus forgetting may be avoided. Our setting is significantly different from theirs and more challenging. We focus on mitigating catastrophic forgetting with very limited access to previous domain data. Other paper citations about memory-based meta learning research papers are added. See the related work discussion marked as “red” part.
>
> Q: How does adaptation of learning rate address catastrophic interference?
>
> A: In previous works, it has been found different portions of a neural network are trained to different extents in the learning [1]. While some portions of a network are fully exploited and their parameters have been tailored towards past learned tasks, other parts can remain less exploited and have bigger potential to learn new tasks. In this work, we proposed this block-level learning rate adaptation mechanism to harness this inherent training dynamic of neural network. The proposed adaptation mechanism is optimized based on the meta loss in memory tasks, which helps the model to learn general domain-level features and neglect task level variances.
>
> Q: Maybe discuss memory and accuracy trade-off?
>
> A:  We have discussed the memory and accuracy trade-off in Table 6.
>
> Q: Some baselines don’t require a replay buffer nor iterative re-training.
>
> A: Yes, you are right, we just want to make comprehensive comparisons for related baselines and also show the benefit of using a replay buffer. So, we include baselines consisting of both memory-based and without memory. We are sorry that we are not sure the meaning of iterative re-training, could you elaborate it more?
>
> Q: provide references on the following statements - Most existing works focus on developing the generalization ability under a single context/domain. Recently, it has been shown that catastrophic forgetting often occurs when transferring a meta-learning model to a new context.
>
> A: We added references for these claims.
>
> Q: The first sentence in Section 2.1 seems to describe an application of meta-learning to few-shot learning rather than meta-learning itself.
>
> A: We revised the first sentence. See the revision of the first sentence about meta learning related work discussion
>
> Q: miniImageNet subset was introduced in Matching Nets
>
> A: We update the citation.

---

### Official Review · AnonReviewer2 · 2020-10-29
**Review of the Paper**

**Rating:** 4
**Confidence:** 5

**Review:**

-Summary-
The paper proposes a method for the sequential meta-learning problem. The author meta learn not only model parameters but also learning rate vectors for parameter blocks. To this end, the meta-learn model finds appropriate model parameters and adaptive learning rate vectors that capture task-general information. Overall experiments are performed on few-shot meta-learning settings with sequential domains (datasets).

-Pros-
- Optimizing fine-grained learnable learning rate vectors for manually grouped parameter blocks is reasonable.
- The performance of the proposed model significantly outperforms baselines which naively combined existing few-shot meta-learning and continual learning approaches.


-Cons-
- The approach is too simple (Adding learnable strength vector weights for gradient update of conv. blocks) and heuristic. And core hyperparameters are manually decided, like # of blocks per Conv. layer and the size of memory.
- Lack of analysis. There is a lack of concrete insight into how does adaptive learning rate weights mitigating forgetting.
- The method only considers multi-head continual learning problems for task-incremental learning. The majority of recent impressing CL works considers further realistic and applicable to the broader areas, called class-incremental learning problem that task oracle isn't given during training/inference of the model.
- The method is only performed on simple CNN architectures. It needs to be validated on further modern deep neural network architectures. And, while the construction of CNN in this paper, most of the well-known CNN architectures have a different number of filters per layer. In this case, the strategy to split blocks can be important for pursuing a better model. However, there is no discussion/analysis of the problem.
- Meta-learning with bilevel optimization might require an additional computational cost.

-Comments-
- Experiments on domain ordering are interesting. I see that recent CL works consider evaluations on multiple domains(datasets) like "Hard Attention on Task" (HAT) paper. It would be interesting to perform analysis of the task(domain)-order sensitivity like 'order-normalized performance disparity' (OPD) in [1], which can be beneficial for understanding backward-forward transfer during continual learning under the domain shift.
- Citation of the XtarNet is duplicated. Please combine them.

[1] Yoon, Jaehong, et al. "Scalable and Order-robust Continual Learning with Additive Parameter Decomposition.", ICLR 2020.

---

> ### Author Response · Authors · 2020-11-23
> **Response to Reviewer 2**
>
> Thanks for your suggestion and comments for improving our paper. We present our response to your concerns.
>
> Q: hyperparameters are manually decided, like # of blocks per Conv. layer and the size of memory.
>
> A:  Hyperparamters for our method are not difficult to decide, as presented in our detailed ablation study,  our methods are not sensitive to these hyperparameters.
>
> Q: Lack of analysis. There is a lack of concrete insight into how does adaptive learning rate weights mitigating forgetting.
>
> A: In previous works, it has been found different portions of a neural network are trained to different extents in the learning [1]. While some portions of a network are fully exploited and their parameters have been tailored towards past learned tasks, other parts can remain less exploited and have bigger potential to learn new tasks. In this work, we proposed this block-level learning rate adaptation mechanism to harness this inherent training dynamic of neural network. The proposed adaptation mechanism is optimized based on the meta loss in memory tasks, which helps the model to learn general domain-level features and neglect task level variances.
>
> [1] Sang-Woo Lee et al. Overcoming Catastrophic Forgetting by Incremental Moment Matching, NIPS 2017
>
>
> Q: verify on modern architecture
>
> A: Meta learning models usually adopt small networks. For large models, without pretraining, the model could easily overfit. Even though some works adopt relative larger architectures in current meta learning models, they are usually pretrained on large datasets with large number of image classes [1]. For our setting, the future domain is unknown. Thus a pretrained feature extractor may violate the meta learning principle as the future domain image classes may have overlap classes with pretrained image classes. When meta testing, the model may already see unseen categories
>
> [1] Meta-Learning with Latent Embedding Optimization
> Andrei A. Rusu, Dushyant Rao, Jakub Sygnowski, Oriol Vinyals, Razvan Pascanu, Simon Osindero, Raia Hadsell
>
> Q:  strategy to split blocks can be important for pursuing a better model.
>
> A:   splitting blocks is not difficult, one can even assign one learnable parameter for each CNN filter. It is not necessary to spend much time to tune this hyperparameter.
>
> Q: computation cost
>
> A: Our methods are much faster than both MER[1] and UCB[2] (at least 3 times faster during the experiment, the former one needs more for loops in each iterations due to reptile algorithm computation, and the later one is slow is due to multiple MCMC sampling to calculate the loss function). Our method has comparable computation cost than other methods.
>
> [1] Learning to Learn without Forgetting by Maximizing Transfer and Minimizing Interference
> Matthew Riemer, Ignacio Cases, Robert Ajemian, Miao Liu, Irina Rish, Yuhai Tu, Gerald Tesauro
>
> [2] Uncertainty-guided Continual Learning with Bayesian Neural Networks
> Sayna Ebrahimi, Mohamed Elhoseiny, Trevor Darrell, Marcus Rohrbach
>
> Q: multi-head, task information not available
>
> A: We are working on this line of research, which is a non-trivial extension of meta learning. Thus we consider this as interesting future work.
>
> Q: Citation of the XtarNet is duplicated and order sensitivity analysis
>
> A: we combined them into one. domain ordering sensitivity analysis will be done in future work.

---

### Author Response · Authors · 2020-11-23
**Summary of Author Response**

We thank all the reviewers for their helpful comments and we made a major revision to our paper. The following is the list of our major revisions. The text that is marked as red is the revision part.

[1] We add one more domain sequence experiment, with the ordering of  Omniglot-Aircraft-CUB-CIFARFS-MiniImagenet.

[2] We add motivated scenarios, for motivating our setting and method.

[3] We rewrite the method and algorithm part to reflect the separate discussion of domain-shared and domain-specific model parameters and learning rates.

[4] We add more citations in the introduction to support the claims, and merge one paragraph in the introduction with related works.

---

### Decision · Program_Chairs · 2021-01-07
**Final Decision**

**Decision:**

Reject

**Comment:**

The paper proposes a sequential meta-learning method over few-shot sequential domains, which meta learns both model parameters and learning rate vectors to capture task-general representations.

Reviewers raised many insightful and constructive comments. The main themes are as follows:
- The problem setting needs further motivation and clarifications, to make it more realistic and applicable.
- The novelty is relatively weak, e.g. the approach is too simple, and learning the learning rate is a common trick.
- The method needs great effort for better presentation and justification. The current presentation simply lists several equations in a dense way without detailed explanation. Some main claims such as mitigating catastrophic forgetting are not elaborated extensively.

AC scanned through the paper and agreed with the reviewers' main points. Authors' rebuttal in general did not address these concerns to the satisfaction. For example, even after revision, the readability of this paper is not good enough. The authors are encouraged to perform a thorough revision.